# Prediction of Response to Cisplatin-Based Neoadjuvant Chemotherapy of Muscle-Invasive Bladder Cancer Patients by Molecular Subtyping including KRT and FGFR Target Gene Assessment

**DOI:** 10.3390/ijms23147898

**Published:** 2022-07-18

**Authors:** Thorsten H. Ecke, Paula Carolin Voß, Thorsten Schlomm, Anja Rabien, Frank Friedersdorff, Dimitri Barski, Thomas Otto, Michael Waldner, Elke Veltrup, Friederike Linden, Roland Hake, Sebastian Eidt, Jenny Roggisch, Axel Heidenreich, Constantin Rieger, Lucas Kastner, Steffen Hallmann, Stefan Koch, Ralph M. Wirtz

**Affiliations:** 1Department of Urology, Helios Hospital Bad Saarow, 15526 Bad Saarow, Germany; steffen.hallmann@helios-gesundheit.de; 2Department of Urology, Charité—Universitätsmedizin Berlin, Corporate Member of Freie Universität Berlin, Humboldt-Universität zu Berlin, and Berlin Institute of Health, 10117 Berlin, Germany; paula-carolin.voss@charite.de (P.C.V.); thorsten.schlomm@charite.de (T.S.); anja.rabien@charite.de (A.R.); f.friedersdorff@keh-berlin.de (F.F.); 3Department of Urology, Evangelisches Krankenhaus Königin Elisabeth Herzberge, 10365 Berlin, Germany; 4Department of Urology, Rheinlandklinikum, 41464 Neuss, Germany; dimitri.barski@rheinlandklinikum.de (D.B.); thomas.otto@rheinlandklinikum.de (T.O.); 5Department of Urology, St. Elisabeth Hospital, 50935 Cologne, Germany; michael.waldner@hohenlind.de; 6STRATIFYER Molecular Pathology GmbH, 50935 Cologne, Germany; elke.veltrup@stratifyer.de (E.V.); friederike.linden@stratifyer.de (F.L.); ralph.wirtz@stratifyer.de (R.M.W.); 7Institute of Pathology, St. Elisabeth Hospital, 50935 Cologne, Germany; roland.hake@hohenlind.de (R.H.); sebastian.eidt@hohenlind.de (S.E.); 8Institute of Pathology, Helios Hospital, 15526 Bad Saarow, Germany; jenny.roggisch@helios-gesundheit.de (J.R.); stefan.koch@helios-gesundheit.de (S.K.); 9Department of Urology, Universitäsklinikum Köln, 50937 Cologne, Germany; axel.heidenreich@uk-koeln.de (A.H.); konstantin.rieger@uk-koeln.de (C.R.); lucas.kastner@uk-koeln.de (L.K.); 10Brandenburg Medical School, 16816 Neuruppin, Germany

**Keywords:** bladder cancer, neoadjuvant chemotherapy, FGFR1, FGFR3, KRT5, KRT20

## Abstract

Patients with muscle-invasive urothelial carcinoma achieving pathological complete response (pCR) upon neoadjuvant chemotherapy (NAC) have improved prognosis. Molecular subtypes of bladder cancer differ markedly regarding sensitivity to cisplatin-based chemotherapy and harbor FGFR treatment targets to various content. The objective of the present study was to evaluate whether preoperative assessment of molecular subtype as well as FGFR target gene expression is predictive for therapeutic outcome—rate of ypT0 status—to justify subsequent prospective validation within the “BladderBRIDGister”. Formalin-fixed paraffin-embedded (FFPE) tissue specimens from transurethral bladder tumor resections (TUR) prior to neoadjuvant chemotherapy and corresponding radical cystectomy samples after chemotherapy of 36 patients were retrospectively collected. RNA from FFPE tissues were extracted by commercial kits, Relative gene expression of subtyping markers (e.g., KRT5, KRT20) and target genes (FGFR1, FGFR3) was analyzed by standardized RT-qPCR systems (STRATIFYER Molecular Pathology GmbH, Cologne). Spearman correlation, Kruskal–Wallis, Mann–Whitney and sensitivity/specificity tests were performed by JMP 9.0.0 (SAS software). The neoadjuvant cohort consisted of 36 patients (median age: 69, male 83% vs. female 17%) with 92% of patients being node-negative during radical cystectomy after 1 to 4 cycles of NAC. When comparing pretreatment with post-treatment samples, the median expression of KRT20 dropped most significantly from DCT 37.38 to 30.65, which compares with a 128-fold decrease. The reduction in gene expression was modest for other luminal marker genes (GATA3 6.8-fold, ERBB2 6.3-fold). In contrast, FGFR1 mRNA expression increased from 33.28 to 35.88 (~6.8-fold increase). Spearman correlation revealed positive association of pretreatment KRT20 mRNA levels with achieving pCR (r = 0.3072: *p* = 0.0684), whereas pretreatment FGFR1 mRNA was associated with resistance to chemotherapy (r = −0.6418: *p* < 0.0001). Hierarchical clustering identified luminal tumors of high KRT20 mRNA expression being associated with high pCR rate (10/16; 63%), while the double-negative subgroup with high FGFR1 expression did not respond with pCR (0/9; 0%). Molecular subtyping distinguishes patients with high probability of response from tumors as resistant to neoadjuvant chemotherapy. Targeting FGFR1 in less-differentiated bladder cancer subgroups may sensitize tumors for adopted treatments or subsequent chemotherapy.

## 1. Introduction

Bladder cancer is still a highly frequent cancer in Europe with an incidence of nearly 200,000 cases and an annual mortality rate of 64,966 cases in 2018 [1]. Approximately 30% of these patients suffer from muscle-invasive bladder cancer (MIBC) at the time of initial diagnosis [2]. Up-to-date radical cystectomy (RC) with lymph node dissection remains the recommended treatment in highest-risk non-muscle-invasive and muscle-invasive nonmetastatic bladder cancer, preceded by cisplatin-based neoadjuvant chemotherapy (NAC) in eligible patients [3].

In order to remedy this unsatisfactory situation, serious efforts have recently focused on new therapeutic strategies regarding the application of neoadjuvant and adjuvant chemotherapies [4]. A better risk assessment of patients has been recommended by developing novel predictive/prognostic models [5]. In clinical practice, the therapeutic management of these patients has so far been performed almost exclusively on the basis of clinical data and classical pathological TNM criteria, but with few reliable results [5]. Neoadjuvant treatment modalities are still not widely accepted due to their remaining inability to accurately select patients who will benefit vs. those who may potentially be harmed [6]. It is hoped that the identification of new molecular tissue biomarkers could help to stratify risk groups and determine patients who could have a benefit from adjuvant strategies after surgery [7].

The molecular subtyping of bladder cancer has been well accepted since its initial introduction in 2014 [8,9,10]. Therein, the quantitation of *KRT5* and *KRT20* on mRNA level and/or their recapitulation on protein level by IHC belong to common-sense hallmarks of molecular subtyping of basal and luminal tumors, respectively. In a previous work, we showed that *KRT20* is strongly associated with adverse outcome for pT1 NMIBC as well as chemotherapy-naïve MIBC [11,12].

Importantly, molecular subtyping of MIBC may also play a role as a potential biomarker for neoadjuvant treatment response. When molecular classification is to be translated into clinical use, it is important to consider that the several classification methods emphasize slightly different aspects of tumor biology [13].

However, the predictive role of these markers in MIBC patients receiving neoadjuvant chemotherapy is still unknown. Early studies indicate a tremendous decrease in KRT20 mRNA levels, when comparing matched TURB and cystectomy tissue samples before vs. after neoadjuvant chemotherapy [14]. The aim of the present study was to evaluate the predictive role of KRT20 in combination with potential, druggable resistance markers in the neoadjuvant situation and to prove their clinical usefulness.

## 2. Results

### 2.1. Distribution of Assessed Protein and mRNA Markers across the Study Cohort

As depicted in the remark diagram (Figure 1), TURB biopsies from 36 patients could be analyzed, with both clinical data and matched tissues being available.

After radical cystectomy, the ypT0 rate was 39%, four patients showed lymph node metastases (11%) and one patient had positive margin (3%). The number of resected lymph nodes ranged from 8 to 30 (average was 14).

All investigated experimental markers could be determined by PCR of urinary bladder cancer TURB biopsies as well as cystectomy tissue. As depicted in Figure 2, the relative gene expression of multiple subtype-specific marker genes significantly differed between TUR biopsy and matched cystectomy specimen. Most prominently, the median mRNA expression of the luminal marker gene KRT20 decreased from 37.76 to 30.65 (138.1-fold), while the decrease was less prominent for other luminal marker genes, such as GATA3 (decrease from 38.81 to 36.20; 6.1-fold) and ERBB2 (decrease from 37.94 to 35.40; 5.8-fold). Interestingly, the median expression level of the basal marker gene KRT5 decreased less substantially after neoadjuvant chemotherapy (decrease from 36.36 to 35.27; 2.1-fold), while a marked change could be observed for the upper quartile of KRT5 mRNA expression (decrease from 40.59 to 36.41; 18.1-fold). In contrast, the median expression of FGFR1 mRNA was higher in cystectomy samples after neoadjuvant chemotherapy compared to matched TUR biopsy samples before therapy (increase from 33.29 to 35.89; 6.1-fold), while its receptor tyrosine kinase family member FGFR3 was significantly lower in cystectomy samples after chemotherapy (decrease from 37.89 to 34.98; 7.5-fold). When comparing the pre-therapy total gene expression data distribution for each marker, with the gene expression data distribution of tumors achieving a pathological complete response, it became apparent that the responding tumors were disproportionally enriched in the high KRT20 expressors and low FGFR1 expressors.

### 2.2. Correlation of mRNA Markers on Basis of Molecular Subtyping and Clinical Variables

As previously described and depicted in Figure 3, the mRNA expression of basal and luminal marker genes was negatively associated in TURB biopsy samples before chemotherapy. KRT5 was negatively associated with the luminal marker’s genes KRT20, GATA3 and ERBB2 (r = −0.6111, *p* < 0.0001; r = −0.4782, *p* = 0.0032 and r = −0.3611, *p* = 0.0305, respectively). However, FGFR3 mRNA expression was positively associated with the dominant luminal marker gene KRT20 (r = 0.3470, *p* = 0.0381), while virtually no association could be detected with the basal marker gene KRT5 (r = 0.0921, *p* = 0.5930).

Importantly, Spearman correlation supported the previously mentioned inverse relation of KRT20 and FGFR1 mRNA expression with pCR status. While KRT20 mRNA tended to be positively associated with pCR status (r = 0.3072; *p* = 0.0684), the negative association of FGFR1 mRNA expression with pCR status was highly significant (r = −0.65418, *p* < 0.0001).

### 2.3. Hierarchical Clustering Defines Subgroup of Chemotherapy Resistant Tumors

The relative mRNA expression of the candidate genes was used to perform two-way hierarchical clustering, and clinical outcome was superimposed to characterize the arising patient groups.

As depicted in Figure 4, hierarchical clustering revealed two KRT5-positive basal clusters with moderate pCR rate (3/11; 27%), one KRT20-positive luminal cluster with high pCR rate (10/16; 63%) and one “double negative” subgroup with both keratins (KRT5 and KRT20) being expressed at very low levels but exhibiting high FGFR1 expression, which we therefore named stromal-rich tumors. The stromal-rich tumor subgroup had low pCR (1/9; 12.5%), with the only exception in the stromal cluster having again high KRT20 and low FGFR1 expression, indicating the limitation of the cluster method. However, in summary, the luminal cluster exhibited a twofold higher pCR rate, while the stromal-rich tumors exhibited a threefold lower pCR rate compared to the overall pCR rate of 38%.

### 2.4. Contingency Testing to Evaluate Predictive Value of Marker Genes

To overcome limitations of the clustering method to predict the outcome of the partitioning method was used to define the optimal cut-off to predict pathological complete response by KRT20 and FGFR1 mRNA levels. When applying these cut-offs in contingency tests, both markers revealed themselves to be predictive for clinical outcome (Figure 5).

Stratification based on KRT20 mRNA did separate tumors exhibiting high KRT20 mRNA expression and high pCR rate (66.7%) from tumors with low KRT20 mRNA expression and low pCR rate (25%). This separation was significant in a chi-squared test (Chi^2^ = 5.845, *p* = 0.0156). Stratification based on FGFR1 mRNA did separate tumors exhibiting high FGFR1 mRNA expression and low pCR rate (0%) from tumors with low FGFR1 mRNA expression and high pCR rate (66.7%). This separation was highly significant in chi-squared testing (Chi^2^ = 21.38, *p* < 0.0001). 

## 3. Discussion

Since the discovery of luminal and basal subtypes in muscle-invasive bladder cancer in 2014, their impact on response to neoadjuvant chemotherapy has been discussed [9]. Interestingly, already in the first report, basal tumors characterized, i.a., by high KRT5 mRNA expression, as well as luminal tumors exhibiting, i.a., high KRT20 mRNA expression had intermediate to high pathological complete response rates ranging between 25 and 60%. In contrast, the so-called “p53-like” tumors, bearing no p53 mutation and exhibiting low KRT5 and KRT20 mRNA expression did not or only marginally responded to upfront chemotherapy. Interestingly, when comparing gene expression levels in TUR biopsies before neoadjuvant chemotherapy with matched cystectomy tissue after treatment, the frequency of luminal tumors dropped, while basal tumors remained similar and “p53-like tumor” increased [9].

However, molecular subtyping evolved and became more complex. Recently, a substratification of the original tripartite molecular subtypes has been published as “consensus molecular classification of muscle invasive bladder cancer” that distinguishes “Luminal Papillary”, “Luminal unstable”, “Luminal unspecified”, “Basal/Squamous”, “Stroma-rich” and “Neuroendocrine-like” subtypes. However, while the diverse subtyping approaches were integrated by quantifying similarities of genome-wide RNAseq-based expression analysis using Cohens kappa scores and constructing clustered networks, the clinical impact and prognostic value became less apparent, with the smallest subtype (“Neuroendocrine-like”) having a markedly different, worse outcome [15]. Still the molecular sub-subtyping provoked subtle differentiation of hypothesized best treatment options, with “Luminal-papillary” and “Luminal-infiltrated” having “low predicted likelihood of response” to a neoadjuvant chemotherapy report [8] in contrast to the initial report [9].

Here, we have used RT-qPCR-based quantitation of predefined hallmark subtyping markers (KRT5/KRT20) [8], with proven prognostic impact in muscle-invasive and non-muscle-invasive bladder cancer [11,12] and which have been shown to have some predictive value in a finding cohort [14]. Moreover, we have integrated FGFR1 and FGFR3mRNA expression analysis into the predictive model to evaluate the impact of stromal interactions and therapeutic implications in view of pan-FGFR inhibitors entering the field.

We could validate a dramatic decrease in luminal marker gene expression as exemplified by KRT20, which is congruent with the initial finding of Choi et al., from matched-pair analysis, in which luminal subtype signatures get lost after neoadjuvant chemotherapy. Moreover, we could recapitulate the finding of an independent previous cohort, where KRT20 also exhibited a dramatic decrease in overall expression [14]. Therefore, we conclude that luminal tumors, defined by high KRT20 mRNA expression, do have a high likelihood of responding to neoadjuvant chemotherapy (66.7% pCR rate in KRT20 high tumors vs. 25% pCR rate in KRT20 low tumors). This is in sharp contrast to previous hypothetical assumptions [8], but in line with the initial original work [9].

Moreover, by analyzing FGFR1 mRNA expression, a stromal-rich tumor subtype has been identified that lacks both KRT5 and KRT20 mRNA expression and is almost resistant to upfront chemotherapy (0% pCR rate in FGFR1 high tumors vs. 66.7% pCR rate in FGFR1 low tumors). This stromal-rich tumor subtype has similarities with the “p53-like” non-responding tumors not responding to upfront chemotherapy in the initial subtyping landmark paper [9], while the major impact of FGFR1 itself has not been reported in previous publications. Importantly, FGFR1is a bona-fide target of FGFR inhibitors, which had initially been introduced into the treatment of metastatic bladder cancer, which harbors FGFR3 mutations or fusions [16]. It is tempting to speculate whether blocking FGFR1 activity in stromal-rich subtypes can restore sensitivity to neoadjuvant chemotherapy in otherwise resistant, muscle-invasive bladder cancer, which warrants further clinical investigation.

As discussed above, Choi et al. [9] identified a basal, a luminal and a so-called p53-like subtype. Approximately one-third of patients belonged to each subtype [9]. They initially reported that p53-like tumors were more resistant to NAC than luminal or basal tumors [9]. Subsequent publications focused on the survival benefit of basal tumors, which in the absence of NAC were associated with the worst prognosis but had the best prognosis after NAC [17]. Recently, Seiler et al. [18] developed a single-sample genomic subtyping classifier based on samples classified according to the molecular subtyping methods of the aforementioned projects. OS and pCR according to subtype (claudin-low, basal, luminal-infiltrated, and luminal) were retrospectively compared for 343 MIBC NAC and 476 MIBC non-NAC cases. Luminal tumors had the longest OS with and without NAC. Nevertheless, OS differed according to the response to NAC. Claudin-low tumors were associated with poor OS irrespective of treatment regimen. Basal tumors showed the highest improvement in OS with NAC compared with surgery alone [18]. Despite having higher case numbers, the analysis lacks the comparison of tumor tissue analysis before and after chemotherapy to conclude on the responsive subtypes. In contrast, the comparison of subtyping markers in TURB versus matched Cystectomy samples is in line with initial and recent publications demonstrating luminal subtype being most strongly affected by chemotherapy in independent cohorts of similar size, as this type of tumor cell is disappearing in the post-treatment samples [9,14].

Furthermore, our findings indicate that luminal tumors defined by high KRT20 mRNA expression do have worse outcomes in MIBC if not treated by (neo)adjuvant chemotherapy, while basal tumors defined by KRT5 mRNA overexpression have better survival irrespective of chemotherapeutic treatment [12]. Of note, in this series, the determination of basal/luminal tumors was performed with an identical molecular test system as has been used in this work. The technique to perform molecular subtyping seems to be critical for prognostic interpretation. While the RNAseq-based subtyping approaches use correlative measures across different platforms against predefined, heterogenous cohorts to vote for a most probable subtype, the RT-qPCR method uses highly sensitive and robust single-marker assessments, with reproducible cut-off values to differentiate between positive and negative marker status on a single sample basis. The same method has been used for outcome prediction and subtyping in breast cancer by developing respective IVD assays [19,20,21,22].

The potential limitations of our study relate to its retrospective design and small cohort size. However, the number of patients was limited; the study included consecutive bladder cancer patients, who were homogeneously being treated with the same neoadjuvant chemotherapy scheme before radical cystectomy at a single center. Because retrospective designs do not guarantee causality, further prospective studies and the use of independent series are warranted to prove the prognostic and predictive value of the analyzed marker combinations to robustly stratify the clinical outcome in real-world assessments, which is the aim of the prospective bladder BRIDGister that had been initiated recently.

Reflecting on the present study in light of already published data, there is reason for optimism that predictive biomarkers will soon be used in clinical practice to guide the use of NAC in patients with MIBC. It seems that, similar to the situation in breast cancer, molecular subtyping of tumors as well as molecular target gene quantification could help to identify tailored treatments in the neoadjuvant setting to optimally address the individual tumor biology of advanced bladder cancer patients. Moreover, applying this approach may help to significantly accelerate the clinical development of new therapeutic options and their optimal sequence with the established chemotherapeutic backbone in defined subtypes of an advanced bladder cancer setting.

It is well-accepted that achieving a pathological complete response after NAC with consecutive RC is associated with improved overall survival [23]. Therefore, both the European and American guidelines recommend a platinum-based neoadjuvant chemotherapy for patients with cT2-T4a cN0cM0 irrespective of molecular subtype [24].

However, most recently it has been shown that efficacy of neoadjuvant chemotherapy is not only important for the immediate tumor regression contributing to an improved survival of patients achieving a pathological complete response, but also a prerequisite for efficacy and survival benefit from subsequent adjuvant checkpoint therapy [25]. In this prospective, randomized clinical trial, the forest plot analysis indicates that adjuvant monotherapy treatment with the anti-PD-1 checkpoint therapy nivolumab was only superior compared to the placebo control arm, when the patients had received a preceding platinum-based neoadjuvant chemotherapy [26]. This suggests that not only patients with pathological complete response towards NAC, but also patients with chemotherapy-sensitive tumors exhibiting minor responses benefit from neoadjuvant chemotherapy, as it forces the remaining tumor tissue to evade the chemotherapy-induced attack by the host’s immune system by manipulating the checkpoint control.

Without preceding chemotherapy, the adjuvant immune therapy is ineffective, as the tumor is masked by the host immune system due to its tumor biology. Based on our findings, we have speculated that particularly luminal tumors with lower immune recognition and subsequent lower immune infiltration, which on the other hand are most sensitive towards chemotherapy, would have the best survival after first/second line checkpoint therapy. Most recently, we could show that indeed KRT20-positive tumors as defined by RT-qPCR from TUR biopsies do have the best survival after second-line anti-PD1 treatment in a retrospective real-world cohort (Wirtz et al. in preparation).

That means that adjuvant immunotherapy is likely to have the greatest impact if its use is guided by predictive biomarkers, selecting the most appropriate neoadjuvant regimen. For tumors not responding to standard chemotherapy as defined by overexpression of stromal signatures and FGFR1 target gene expression, the inhibition of FGFR activities by targeted approaches may be superior to predispose muscle-invasive bladder cancer to subsequent immune oncology treatment. In summary, molecular subtypes and precise target gene assessment on the basis of the underlying tumor biology, as exemplified in this study, seem to be promising to better select the appropriate therapy sequence of standard and upcoming targeted therapies.

## 4. Materials and Methods

### 4.1. Patients

#### Patient Population

From June 2014 to March 2021, a total of 55 patients were included in the trial. After evaluating the necessary data set and FFPE tissue, in total, 36 cases could be included. Representative tissue from the primary tumor as well as from cystectomy tissue was mandatory. Together, 30 male patients and 6 female patients (average age 69 years, range 53–85 years) were included. Pathohistological T-category and grade for the primary tumors are as follows: The study included for the primary tumors pTaG2 (n = 2), pT1G2 (n = 3), pT1G3 (n = 5), pT2G2 (n = 3), and pT2G3 (n = 23) obtained by transurethral resection under institutional-review-board-approved protocols. Three patients showed carcinoma in situ (8%). All non-muscle invasive urothelial carcinomas included in the study progressed to muscle-invasive tumors under the follow-up. All patients were treated with radical surgery after receiving neoadjuvant chemotherapy. Patient characteristics including clinical lymph node status before chemotherapy as well as ECOG performance status at the point of starting chemotherapy are summarized in Table 1.

### 4.2. Eligibility

Eligible patients for this trial were required to have histologically confirmed MIBC transitional cell carcinoma in the bladder. Patients who had received a previous systemic chemotherapy regimen were excluded. Previous radiation therapy was also an exclusion criterion.

Additional eligibility requirements included the following: an ECOG performance status of 0 to 2, a leukocyte count ≥3000/µL, a platelet count ≥100,000/µL, serum bilirubin <1.5 mg/dL, glomerular filtration rate >60 mL/min, and age >18 years. Patients with other active malignancies or any other serious or active medical conditions were excluded. Pregnant or lactating females were ineligible. All patients were required to provide written informed consent prior to the study enrolment.

### 4.3. Pretreatment Evaluation

Prior to enrolment in this trial, all patients were required to have a complete history, physical examination, complete blood counts, chemistry profile, and urine analysis. In addition, patients underwent computed tomography scans of the chest, abdomen, and pelvis with appropriate tumor measurements.

### 4.4. Assessment of Treatment Efficacy

All patients received treatment with the following regimen: gemcitabine at a dose of 1000 mg/m^2^ as a 30 min intravenous infusion on days 1 and 8. On day 2, cisplatin at a dose of 70 mg/m^2^ was administered as an intravenous infusion and hydration with 2000 mL NaCl 0.9%. The regimen was repeated every 21 days. Patients received standard premedication and antiemetic prophylaxis. Patients were evaluated for response to treatment after the completion of 2 courses (6 weeks). Reevaluation included a repeat of all previously abnormal radiologic studies with repeat of objective tumor measurement. Patients received 1 to 4 (median 2) cycles of NAC.

### 4.5. Dose Modifications

All patients received full doses of both agents on day 1 and 2 of the first course of treatment. Subsequent doses were based on hematologic and non-hematologic toxicity observed. Dose modifications for myelosuppression were determined by the blood counts measured on the day of scheduled treatment. Nadir blood counts were not used as a basis for dose reduction.

On day 1 of each course, full doses of all drugs were administered if the leukocyte count was ≥3000/µL and the platelet count was >100,000/µL. If the leukocyte count was <3000/µL or the platelet count was <100,000/µL, patients received granulocyte colony stimulating factor.

### 4.6. Criteria for Follow-Up

The follow-up consisted of clinical examination, ultrasound of abdomen and computed tomography scans of the chest, abdomen, and pelvis with appropriate tumor measurements every 6 months. Progression was defined as new metastatic disease or local progress during follow-up. Chemotherapy response was defined as absence of recurrence, progression, or death from the disease during follow-up. Responses were defined using the response evaluation criteria in solid tumors (RECIST). A complete response (CR) after neoadjuvant chemotherapy was defined as ypT0 in final histopathological report after cystectomy.

### 4.7. Surgical Intervention

All urinary diversions were performed as open surgeries by one surgeon who had more than 10 years operative experience in practice after fellowship. Men underwent removal of the prostate if present and women underwent hysterectomy and bilateral salpingo-oophorectomy if those organs were present. The extent of the pelvic lymph node dissection (PLND) was left to the discretion of the surgeon based on clinician preference and judgment (extent of disease, vascular disease). The extent of PLND was alterable intraoperatively based on clinical findings (vascular disease, fibrosis, adenopathy). After completion of radical cystectomy plus PLND, the open urinary diversion was performed based on preoperative and intraoperative assessments and previous patient discussion.

### 4.8. Isolation of Tumor RNA

After histopathological confirmation of >20% tumor content in TUR biopsy samples based on HE stain evaluation, one subsequent 10 µm slice was used for RNA extraction from FFPE tissue with a commercially available bead-based extraction method (XTRACT kit; STRATIFYER Molecular Pathology GmbH, Cologne, Germany). Similarly, for cystectomy, representative tumor blocks with sufficient tumor content were histopathologically selected for RNA extraction. In cases of pathological complete response, representative scar tissue indicative of former presence of tumor cells was selected as comparative control tissue. After binding and washing to magnetic beads, the RNA was eluted with 100 μL elution buffer and RNA eluates were then stored at −80 °C until use.

### 4.9. Gene Expression by RT-qPCR

The mRNA expression levels of *KRT5*, *KRT20*, *ERBB2*, *GATA3*, *FGFR3* and *FGFR1*, as well as one reference gene (REF), namely *CALM2*, were determined by RT-qPCR, which involves reverse transcription of RNA and subsequent amplification of cDNA executed successively as a 1-step reaction using inventoried validated TaqMan Gene Expression Assays (MP002, MP015, MP452, MP689, MP599 and MP597, STRATIFYER Molecular Pathology GmbH, Köln, Germany). The robustness and usefulness of *CALM2* as a housekeeping gene for diverse candidate genes as well as comparability to diverse IHC assessments such as CK20/KRT20, MKI67/Ki67 and PDL1, when used as single reference gene, has been demonstrated in multiple publications [11,26,27] and resulted in the introduction of *CALM2* as the housekeeping gene in CE-certified IVD products such as Endopredict [19] and MammaTyper [28]. Each patient sample or control was analyzed with each assay mix in triplicate. The experiments were run on a Siemens Versant (Siemens, Erlangen, Germany) according to the following protocol: 5 min at 50 °C and 20 s at 95 °C, followed by 40 cycles of 15 s at 95 °C and 60 s at 60 °C. Forty amplification cycles were applied, and the cycle quantification threshold (Cq) values of three markers and one reference gene for each sample (S) were estimated as the median of the triplicate measurements. The final values were generated by subtracting the CT levels from the reference gene CALM2 from the CT level of the candidate gene to result in Delta CT values (DCT). The DCT was subsequently subtracted from the total number of cycles (40-DCT) to ensure that normalized gene expression obtained by the test was proportional to the corresponding mRNA expression levels, and higher 40-DCT values mean higher mRNA expression levels.

### 4.10. Statistical Analysis

Medians and interquartile ranges (IQR) were determined for continuous variables as well as frequencies and proportions for categorical variables, and candidate gene mRNA expression levels were plotted as data distributions with 40-DCT values on the y-axis. Correlation analyses were performed using Spearman rank correlations. Partition models were generated to create contingency tables with optimal cut-offs. Two-way hierarchical clustering using the continuous mRNA expression values of KRT5, KRT20, GATA3, ERBB2, FGFR3 and FGFR1 was performed using Ward’s minimum variance method, wherein the distance between two clusters is the ANOVA sum of squares between the two clusters added up over all the variables. Ward’s method joins clusters to maximize the likelihood at each level of the hierarchy under the assumptions of multi-variate normal mixtures, spherical covariance matrices, and equal sampling probabilities. Finally, nonparametric testing and a chi2 test were conducted to examine differences in continuous and categorical variables as appropriate. All statistical tests were two-sided and *p*-values < 0.05 were considered statistically significant. All tests and calculations were performed using the software R, version 3.1.2 (R Development Core Team 2014) or JMP 9.0.0 (SAS Institute Inc., 100 SAS Campus Drive Cary, Cary, NC 27513-2414, USA).

### 4.11. Ethics

The study was performed according to the Declaration of Helsinki. The study was approved by the local Institutional Review Board of National Medical Association Brandenburg (No. AS S19(bB)/2020 dated 4 June 2020). Written informed consent was obtained from each participant.

## 5. Conclusions

Molecular subtyping distinguishes patients with a high probability of response from tumors as resistant to neoadjuvant chemotherapy. Targeting FGFR1 in less-differentiated bladder cancer subgroups may sensitize tumors for adopted treatments or subsequent chemotherapy.

## Figures and Tables

**Figure 1 ijms-23-07898-f001:**
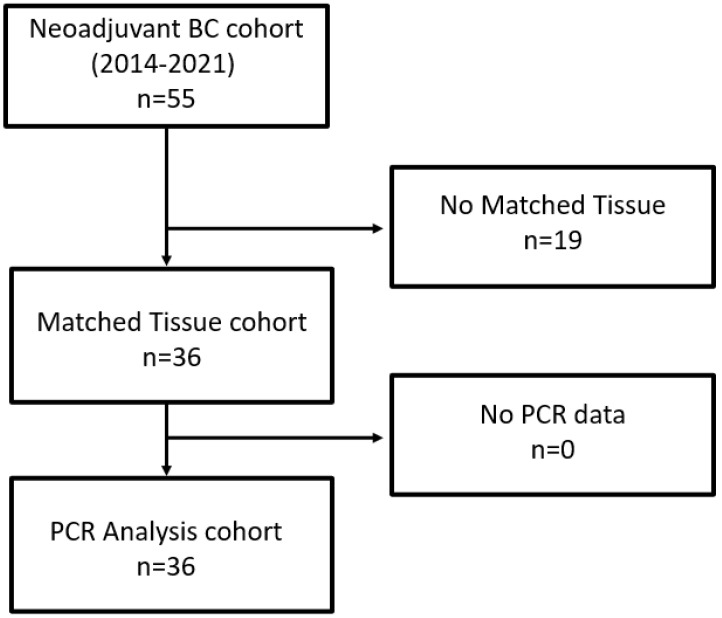
Remark diagram.

**Figure 2 ijms-23-07898-f002:**
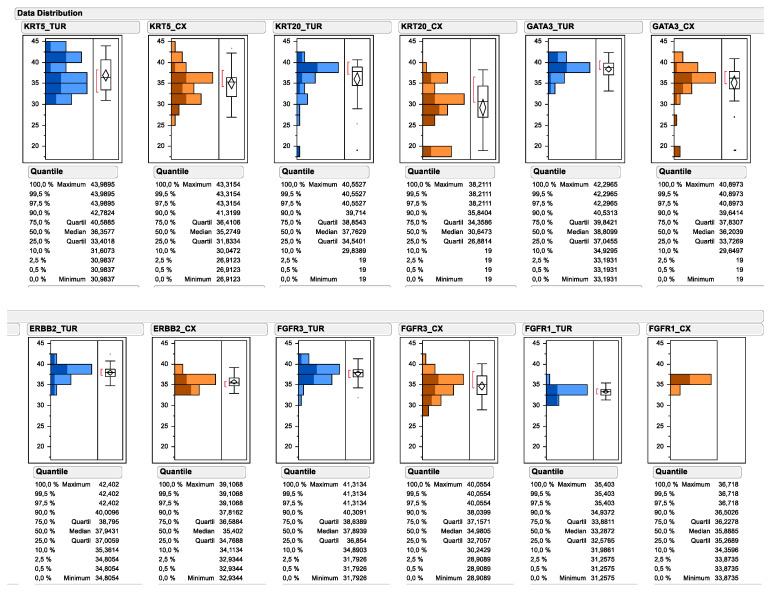
Data distribution of subtyping and target gene expression as determined by standardized RT-qPCR. Continuous mRNA expression levels are depicted as 40-DCT values by subtracting reference gene CT values from candidate gene CT values (=DCT). The subtraction of the DCT from the total number of the PCR reaction converts the numbers, so that higher numbers mean higher expression levels. Pretreatment mRNA expression in TUR biopsies is depicted in blue. Posttreatment mRNA expression is depicted in orange. Upper panel depicts the subtyping marker, lower panel the assessed target genes. Tumor gene expression from tumors achieving pCR are displayed by darker coloring.

**Figure 3 ijms-23-07898-f003:**
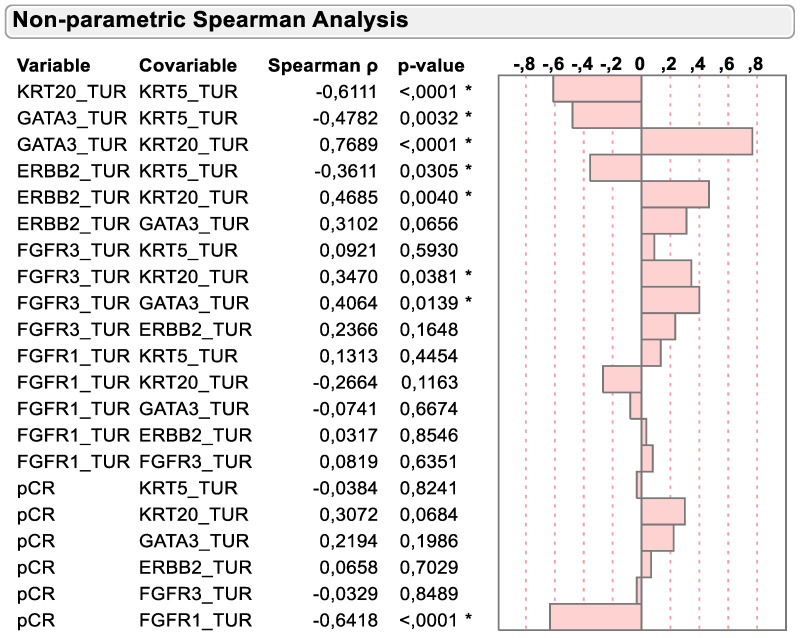
Spearman correlation of KRT5, KRT20, GATA3, ERBB2, FGFR3 and FGFR1 mRNA with dichotomized pCR status. * *p* < 0.05.

**Figure 4 ijms-23-07898-f004:**
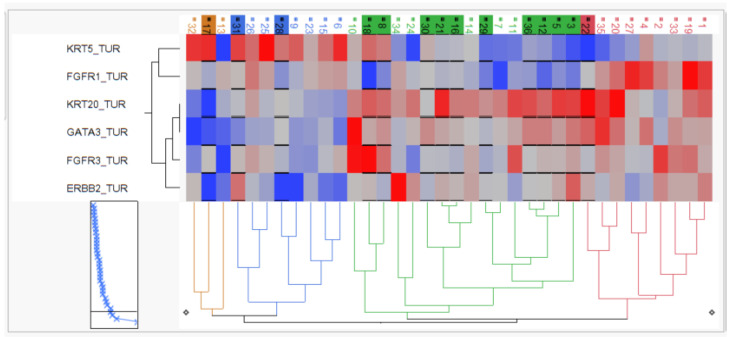
Two-dimensional hierarchy based on KRT5, KRT20, GATA3, ERBB2, FGFR3 and FGFR1 mRNA. Tumors achieving pCR are depicted by black underline and background color of sample IDs.

**Figure 5 ijms-23-07898-f005:**
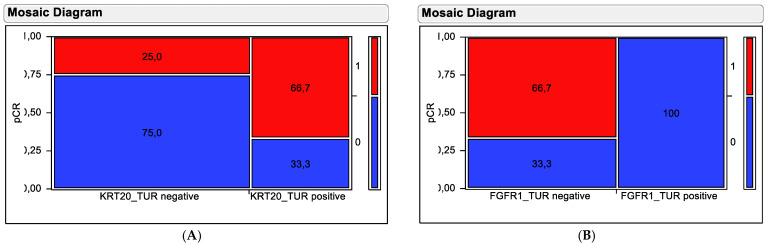
Contingency testing using (**A**) KRT20 mRNA and (**B**) FGFR1 mRNA in pre-therapy TUR biopsy samples for distinguishing responding from non-responding tumors. Tumors achieving pCR are depicted by red fields and resistant tumors are depicted by blue fields. Percentages of tumors in the stratification groups (high versus low expressors) are shown in the fields. The numbers on the left y-axis indicate the proportion in relation to all tumors, while the numbers on the right y-axis indicate the pCR categorization of the coloring with (1 = pCR achieved; 0 = no pCR).

**Table 1 ijms-23-07898-t001:** Clinical characteristics of patients in the total cohort (n = 36).

Cohort	Total Cohort
Size (n)	36
Age (years)	
Average	69
Range	53–85
Gender	
Male	30 (83%)
Female	6 (17%)
ECOG performance status	
0	28 (78%)
1	8 (22%)
2	0 (0%)
Lymph node metastases before chemotherapy	
cN0	29 (81%)
cN1	5 (14%)
cN2	2 (5%)
Response to chemotherapy	
Complete response (ypT0)	14 (39%)
lymph status (ypN0)	32 (89%)

## Data Availability

Not applicable.

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
