# Peer review of "Prediction of Response to Cisplatin-Based Neoadjuvant Chemotherapy of Muscle-Invasive Bladder Cancer Patients by Molecular Subtyping including KRT and FGFR Target Gene Assessment"

_ijms, 2022, doi:10.3390/ijms23147898_

Round 1

Reviewer 1 Report

The work concerns the prediction of the response to neoadjuvant chemotherapy of patients with muscle-invasive bladder cancer by molecular subtyping including evaluation of the target genes KRT5, KRT20 and FGFR.

This is, in fact, a very important topic, with great interest for clinical translation, as there is an increasing need to stratify patients based on genetic/molecular assessment.

The text is well written, clear, but the overuse of abbreviations makes it difficult to follow. Concerning abbreviations, once authors choose to present a listing, they should put them all there. Therefore, the authors must review and add the missing ones.

The introduction well contextualizes the topic, although the authors could have used more recent data (GLOBOCA 2020 data are available).

The materials and methods and the results are well described, with enough detail and the discussion is adequate, allowing a good understanding of the clinical interest.

However, there are some details that need to be improved:

1. throughout the text the unit liter appears as "l" and as "L". authors must harmonize the way of writing throughout the text.

2. Materials and Methods Section: the authors report that of the 55 patients selected, only 36 were included. It is not clear, therefore, why in subsection 2.4,  55 patients are again mentioned. Authors should clarify.

3. Results section: on line 10 on page 6, the values presented for ERBB2 are the same as for GATA3 and do not agree with what is shown in the graphs in figure 2. The authors must correct them.

In subsection 3.2 the authors refer to "...TURB biopsy samples...". It shouldn't just be "...TUR biopsy samples..."'?

3. The text needs a thorough review as there are several places where words appear to be missing.

Author Response

The work concerns the prediction of the response to neoadjuvant chemotherapy of patients with muscle-invasive bladder cancer by molecular subtyping including evaluation of the target genes KRT5, KRT20 and FGFR.

This is, in fact, a very important topic, with great interest for clinical translation, as there is an increasing need to stratify patients based on genetic/molecular assessment.

The text is well written, clear, but the overuse of abbreviations makes it difficult to follow. Concerning abbreviations, once authors choose to present a listing, they should put them all there. Therefore, the authors must review and add the missing ones.

The introduction well contextualizes the topic, although the authors could have used more recent data (GLOBOCA 2020 data are available).

The materials and methods and the results are well described, with enough detail and the discussion is adequate, allowing a good understanding of the clinical interest.

However, there are some details that need to be improved:

  1. throughout the text the unit liter appears as "l" and as "L". authors must harmonize the way of writing throughout the text.

ANSWER: Thanks a lot for that comment. We made all changes following the reviewer’s advice.

  1. Materials and Methods Section: the authors report that of the 55 patients selected, only 36 were included. It is not clear, therefore, why in subsection 2.4,  55 patients are again mentioned. Authors should clarify.

ANSWER: Thanks for the comment. We made the needed change.

  1. Results section: on line 10 on page 6, the values presented for ERBB2 are the same as for GATA3 and do not agree with what is shown in the graphs in figure 2. The authors must correct them.

In subsection 3.2 the authors refer to "...TURB biopsy samples...". It shouldn't just be "...TUR biopsy samples..."'?

ANSWER: Correct, this is a doubling … all "TURB biopsy" have been changed to "TUR biopsy.

  1. The text needs a thorough review as there are several places where words appear to be missing.

ANSWER: We checked the manuscript again carefully and made corrections.

Reviewer 2 Report

The manuscript uses molecular subtyping as a tool to identify patients with a higher probability of response from tumors resistant to neoadjuvant chemotherapy. The overall goal of the study was to evaluate whether the preoperative assessment of molecular subtype and FGFR target gene expression is predictive of therapeutic outcome. Interestingly, FGFR1 in less differentiated bladder cancer subgroups is found to be a target that could sensitize tumors for adopted treatments or subsequent chemotherapy. The novelty of the study lies in the group of patients from which the tissue samples were collected and the target gene that is analyzed in this specific form of cancer. This study could open doors to testing FGFR inhibitors that are currently being developed and offers hope for the treatment of patients.

Author Response

The manuscript uses molecular subtyping as a tool to identify patients with a higher probability of response from tumors resistant to neoadjuvant chemotherapy. The overall goal of the study was to evaluate whether the preoperative assessment of molecular subtype and FGFR target gene expression is predictive of therapeutic outcome. Interestingly, FGFR1 in less differentiated bladder cancer subgroups is found to be a target that could sensitize tumors for adopted treatments or subsequent chemotherapy. The novelty of the study lies in the group of patients from which the tissue samples were collected and the target gene that is analyzed in this specific form of cancer. This study could open doors to testing FGFR inhibitors that are currently being developed and offers hope for the treatment of patients.

ANSWER: We are thankful for the positive comments to our study.

Reviewer 3 Report

In this study, the authors investigated the predictive role of KRT20 in combination with potential, drug resistance markers in the neoadjuvant situation and to prove their clinical usefulness. The authors concluded that molecular subtyping distinguishes patients having high probability of response from tumours being resistant to neoadjuvant chemotherapy; targeting FGFR1 in less differentiated bladder cancer subgroups may sensitize tumours for adopted treatments or subsequent chemotherapy.

Comments:

The reviewer has some concerns as follows:

1. One of the major concerns is that the sample size is too small (total n=36 containing complete response to chemotherapy n=14), which may affect the validity of the conclusions of this study due to limited data.

2. In Figure 1 for remark diagram, the information for surgery and pathological evaluation and the numbers for responders and non-responders can be added.

3. In Figure 2 for data distribution of subtyping and target gene expression, the analytical method and data presentation can be described in detail in the Methods and Results, respectively.

4. In Figure 4 for two dimensional hierarchical based on KRT5, KRT20, GATA3, ERBB2, FGFR3 and FGFR1 mRNA, the analytical method and data presentation can be described in detail in the Methods and Results, respectively. Moreover, the definitions for “high pCR rate” and “double negative” should be clearly described.

5. The data of protein expression for these signaling molecules such as IHC data can strength the evidence.

6. In Figure 5, the numbers on the y-axis are obscured in left and right panels.

7. In Abbreviations list and their in text, the “N” for lymph node status and “M” for metastases status are really inappropriate.

Author Response

In this study, the authors investigated the predictive role of KRT20 in combination with potential, drug resistance markers in the neoadjuvant situation and to prove their clinical usefulness. The authors concluded that molecular subtyping distinguishes patients having high probability of response from tumours being resistant to neoadjuvant chemotherapy; targeting FGFR1 in less differentiated bladder cancer subgroups may sensitize tumours for adopted treatments or subsequent chemotherapy.

Comments:

The reviewer has some concerns as follows:

  1. One of the major concerns is that the sample size is too small (total n=36 containing complete response to chemotherapy n=14), which may affect the validity of the conclusions of this study due to limited data.

ANSWER: Rare application of neo-adjuvant chemotherapy and highly selected population are the reasons for the small case number and exploratory phase I biomarker study, which is similiar sized as the NAC cohort (n=34) in the initial subtyping publication evaluating response to neoadjuvant chemo from the MD Anderson group(Choi et al. Cell 2014).

  1. In Figure 1 for remark diagram, the information for surgery and pathological evaluation and the numbers for responders and non-responders can be added.

ANSWER: Thanks a lot for this helpful comment. We discussed changes according to the reviewer’s advice. We have these informations included in the manuscript, maybe in the remark diagram it might be overloaded.

  1. In Figure 2 for data distribution of subtyping and target gene expression, the analytical method and data presentation can be described in detail in the Methods and Results, respectively.

ANSWER: Thanks a lot for this comment. We followed the advice and made the needed changes in the manuscript.

  1. In Figure 4 for two dimensional hierarchical based on KRT5, KRT20, GATA3, ERBB2, FGFR3 and FGFR1 mRNA, the analytical method and data presentation can be described in detail in the Methods and Results, respectively. Moreover, the definitions for “high pCR rate” and “double negative” should be clearly described.

ANSWER: Thanks a lot for this comment. We followed the advice and made the needed changes in the manuscript.

  1. The data of protein expression for these signaling molecules such as IHC data can strength the evidence.

ANSWER: The comparison of the applied RT-qPCR assays ghas been previously compared to IHC methods indicating with good concordance but better prognostic power and higher precision due to broader dynamic range (Breyer et al Virchows Archive 2017). Due to limited tissue material of the TUR biopsies the IHC staining of all markers was not done in this series.

  1. In Figure 5, the numbers on the y-axis are obscured in left and right panels.

ANSWER: Figure legend amended to clarify the numeration.

  1. In Abbreviations list and their in text, the “N” for lymph node status and “M” for metastases status are really inappropriate.

ANSWER: Thanks a lot for this comment. As this might be self explaning we excluded this points from abbreviation list.

Round 2

Reviewer 3 Report

No further comments. This revised manuscript can be accepted.